# Long-Term Study on Therapeutic Strategy for Treatment of Eisenmenger Syndrome Patients: A Case Series Study

**DOI:** 10.3390/children9081217

**Published:** 2022-08-12

**Authors:** Yi-Ching Liu, Yu-Wen Chen, I-Chen Chen, Yen-Hsien Wu, Shih-Hsing Lo, Jui-Sheng Hsu, Jong-Hau Hsu, Bin-Nan Wu, Yi-Fang Cheng, Zen-Kong Dai

**Affiliations:** 1Department of Pediatrics, Kaohsiung Medical University Hospital, Kaohsiung 80756, Taiwan; 2Department of Nuclear Medicine, Kaohsiung Medical University, Kaohsiung 80756, Taiwan; 3Department of Pediatrics, School of Medicine, College of Medicine, Kaohsiung Medical University, Kaohsiung 80756, Taiwan; 4Graduate Institute of Medicine, College of Medicine, Kaohsiung Medical University, Kaohsiung 80756, Taiwan; 5Department of Radiology, Kaohsiung Medical University, Kaohsiung 80756, Taiwan; 6Department of Pharmacology, School of Medicine, College of Medicine, Kaohsiung Medical University, Kaohsiung 80756, Taiwan

**Keywords:** eisenmenger syndrome, congenital heart disease, pulmonary arterial hypertension, sildenafil, bosentan, positron emission tomography, pulmonary scan

## Abstract

Eisenmenger syndrome (ES) refers to congenital heart diseases (CHD) with reversal flow associated with increased pulmonary pressure and irreversible pulmonary vascular remodeling. Previous reports showed limited therapeutic strategies in ES. In this study, 5 ES patients (2 males and 3 females), who had been followed regularly at our institution from 2010 to 2019, were retrospectively reviewed. We adopted an add-on combination of sildenafil, bosentan, and iloprost and collected the clinical characteristics and outcomes as well as findings of echocardiography, computed tomography, pulmonary perfusion-ventilation scans, positron emission tomography, and biomarkers. The age of diagnosis in these ES patients ranged from 23 to 54 years (38.2 ± 11.1 years; mean ± standard deviation), and they were followed for 7 to 17 years. Their mean pulmonary arterial pressure and pulmonary vascular resistance index were 56.4 ± 11.3 mmHg and 24.7 ± 8.5 WU.m^2^, respectively. Intrapulmonary arterial thrombosis was found in 4 patients, ischemic stroke was noted in 2 patients, and increased glucose uptake of the right ventricle was observed in 4 patients. No patient mortality was seen within 5 years of follow-up. Subsequently, 2 patients died of right ventricular failure, 1 died of sepsis related to brain abscess, and another died of sudden death. The life span of these patients was 44–62 years. Although these patients showed longer survival, the beneficial data on specific-target pharmacologic interventions in ES is still preliminary. Thus, larger trials are warranted, and the study of cardiac remodeling in ES from various CHD should be explored.

## 1. Introduction

Eisenmenger syndrome (ES) is a long-term complication of an unrepaired, congenital heart defect that causes blood to circulate abnormally in the heart and lungs. It is the most severe form of pulmonary arterial hypertension (PAH) related to congenital heart disease (CHD) [1]. Clinically, ES affects multiple organs, resulting in progressive deterioration of their functions over time. Most patients with ES have reduced life expectancy and only survived to the third or fourth decades of their life [2,3].

The signs and symptoms of ES in advanced stages include central cyanosis, dyspnea, fatigue, dizziness, hemoptysis, syncope, and reduced quality and expectancy of life [4]. Clinical management has traditionally focused on supportive treatments, while medical treatments for these patients often included diuretics, digoxin and antiarrhythmic agents. However, none of these approaches has proven to significantly modify the quality of life and survival rate for ES patients [5]. Although heart-lung transplantation has been shown to be effective for the treatment of ES, both the lengthy duration of waiting for heart-lung transplantation along with strict criteria for patient selection have limited its applicability [6]. In addition, the morbidity and mortality after transplantation remained substantially high [7].

Currently, increased understanding of the pathophysiology of ES together with the reported success of disease-specific treatment for pulmonary arterial hypertension (PAH) using advanced therapies (ATs), including phosphodiesterase type 5 inhibitors, endothelin receptor antagonists, and prostacyclin *analogs*, have offered new hope for patients with ES. These therapies for PAH have been shown to improve patients’ functional capacity, quality of life, and long-term survival [8,9]. By adopting similar strategies for the treatment of ES, our institution has now seen a significant improvement in survival for these patients to 50–60 years. Herein, we report our case studies for the treatment of ES patients.

## 2. Materials and Methods

This retrospective and descriptive study included individuals diagnosed with ES who were treated and followed at the Pediatric Cardiopulmonary Department of Kaohsiung Medical University Hospital between January 2010 and December 2019. Diagnosis of ES was based on clinical presentation and echocardiography. The protocol was approved by the Ethics in Research Committee of the institution where treatments were conducted (KMUHIRB-E(I)-20200267). All medical records of the enrolled patients were reviewed retrospectively. Demographic data, including age, sex, age of disease onset, functional class and medications, were collected. Results of laboratory exams including brain natriuretic peptide (BNP) and uric acid, electrocardiography, transthoracic echocardiography (TTE), pulmonary perfusion-ventilation (V/Q) scan, six-minute walk test (6MWT) were obtained for evaluation. Additionally, cardiac catheterization with hemodynamic studies, 2-deoxy-2-(^18^F)fluoro-D-glucose (FDG) positron emission tomography (PET), and pulmonary perfusion-ventilation scans were also analyzed.

## 3. Results

Five adults (two males and three females) were diagnosed with ES secondary to CHD, and they were treated and followed in our institution between January 2010 and December 2019. There were three patients with secundum-type atrial septal defect (ASD) and one patient each with primum-type ASD and a perimembranous ventricular septal defect (VSD). The clinical characteristics of these patients are summarized in Table 1 and Figure 1, Figure 2 and Figure 3. The duration of follow-up ranged from 7 to 17 years (11.6 ± 4.1 years), as several ES patients were diagnosed with other diseases and were already subjected to follow-up studies prior to January 2010. During the long-term follow-up, ES patients were treated with a sequential combination of ATs, including sildenafil, bosentan, and inhaled iloprost.

Chest X-ray records of these patients usually exhibited cardiomegaly and pulmonary trunk engorgement (Figure 4A). Dilatation of the right atrium and right ventricle (RV) was noted by TTE (Figure 4B). In addition, a high incidence of thromboembolic events in these patients was also observed. Furthermore, two cases of ischemic cerebral stroke were seen in brain CT, and five cases of intrapulmonary thrombus were found. Despite that, the location and severity of intrapulmonary thrombus demonstrated in CT (Figure 4C) were not compatible with pulmonary V/Q scan, which disclosed multiple defects (Figure 4D,E). Hemodynamic studies disclosed that the mean pulmonary arterial pressure was 56.4 ± 11.3 mmHg (mean ± standard deviation), and the pulmonary vascular resistance (PVR) index was 24.7 ± 8.5 WU.m^2^. Four patients received PET/computed scans, and they all showed increases in both the uptake of (18F)-FDG in RV as well as the ratio of RV to the left ventricle (LV) (Figure 4F). Finally, five patients died during the follow-up. Their life span was 44–62 years (53.3 ± 8.2 years). A brief history of each patient is summarized below.

### 3.1. Case 1

This male with perimembranous VSD was diagnosed as having ES at 54 years of age. Initially, he received sildenafil. Due to poor responses, bosentan was subsequently added on. Allopurinol and colchicine were administered after the 4th year of follow-up due to hyperuricemia. Two years later, TTE and chest CT both disclosed thrombosis on his proximal right pulmonary artery (RPA), so warfarin was administered since then. This patient had an episode of ischemic stroke of the right thalamus in the 7th year of follow-up. It should be noted that during the entire 8 years of follow-up, his estimated systolic pulmonary arterial pressure (SPAP) by TTE progressed to 110–130 mmHg. A pulmonary scan also revealed perfusion reduction in the left lower lobe. Since the beginning of the 8th year of follow-up, the levels of BNP raised above 300 ng/L. In addition, the 6MWT was always less than 165 m. The oxygen saturation (SpO2) was around 80–85%. The patient died due to sudden death during the 8th year of follow-up.

### 3.2. Case 2

This female with secundum-type ASD was diagnosed as having ES at 41 years of age. Initially, she received sildenafil. Due to poor responses, bosentan and inhaled iloprost were added on subsequently. During the 17-year follow-up, the estimated SPAP by TTE progressed gradually to 110–130 mmHg. Tricuspid annular plane systolic excursion (TAPSE) was around 15 to 20 mm and below 15 mm in the last follow-up year. The last right heart catheterization (RHC) performed 3 years before death disclosed high PVR (17.9 WU) and PVR index (29.7 WU.m^2^). TTE and chest CT both confirmed thrombosis on proximal RPA and proximal left pulmonary artery (LPA) despite being treated with warfarin. Consistent with these findings, a pulmonary perfusion scan discovered defects over the left lower lobe in the anterior view (Figure 4D) and the right middle lobe in the posterior view (Figure 4E). Colchicine was administered during the 7th year of follow-up due to hyperuricemia. Atrial fibrillation was noted in the 10th year, so amiodarone was administered. Despite treatments with sildenafil, bosentan, and inhaled iloprost, her clinical conditions deteriorated gradually. The SpO2 was around 85–90%. The 6MWT declined from about 350 m to around 150 m gradually. The levels of BNP were usually above 300 ng/L during the 12th year of follow-up. We tried to replace inhaled iloprost with intravenous epoprostenol about one month before her death. However, the dosage could not be titrated up due to the adverse effects. The patient died due to RV failure during the 17th year of follow-up.

### 3.3. Case 3

This female with primum-type ASD was diagnosed as having ES at 36 years of age. Initially, she received sildenafil, and her clinical condition improved. The 6MWT was around 350 to 400 m. Her serum uric acid level was normal during the follow-up. A pulmonary scan demonstrated irregular perfusion/ventilation distribution in the bilateral lung fields, asymmetrical lobar reduction of V/Q match defect in the right upper lobe, and perfusion reduction in the right lung. However, clinical deterioration was noted during the 8th year of follow-up. The 6MWT declined below 300 m. Additionally, the levels of BNP elevated to above 300 ng/L. As a result, bosentan and inhaled iloprost were added. Although the 6MWT improved to around 350 m, the levels of BNP were still above 300 ng/L. SpO2 was around 75–80%. The estimated SPAP by TTE was around 70–90 mmHg, and the TAPSE around 15–20 mm. The last RHC undertaken in the 12th year of follow-up disclosed a high PVR (15.3 WU) and PVR index (18.81 WU.m^2^). In addition, atrial fibrillation developed during the 12th year of follow-up, and amiodarone was administered. TTE and chest CT both showed thrombosis on proximal RPA while warfarin was placed. Unfortunately, 2 episodes of ischemic stroke occurred during follow-up (Figure 5). The patient died due to sepsis related to a brain abscess during the 13th year of follow-up.

### 3.4. Case 4

This female with secundum-type ASD status post-surgical repair was diagnosed with ES at 37 years of age. Initially, she received bosentan due to compassionate use, and her clinical condition improved. The 6MWT was around 400 to 450 m. However, clinical deterioration was noted during the 4th year of follow-up. The 6MWT declined to below 300 m. In addition, the levels of BNP significantly elevated to above 1000 ng/L. As a result, sildenafil and inhaled iloprost were also given. Although the 6MWT improved to 300–350 m, the levels of BNP remained above 1000 ng/L. The estimated SPAP by TTE was around 90–110 mmHg. Multiple ill-defined mismatched perfusion defects were noted in the left lung field; neither intracardiac thrombosis nor pulmonary thrombosis was observed by image study during the follow-up. In addition, no episode of stroke occurred in the patient. Elevated serum uric acid levels were measured during the 7th year of follow-up, so colchicine was administered. Massive ascites and pericardial effusion were found in the same year. Her clinical condition deteriorated rapidly in the last half-year of her life. The 6MWT declined to below 100 m. The SpO2 declined to 75–80%. Because of poor quality of life, the patient refused to accept our suggestion to replace the inhaled iloprost with intravenous epoprostenol. She died due to RV failure during the 7th year of follow-up.

### 3.5. Case 5

This male with secundum-type ASD had received surgical repair and was diagnosed as having ES at 23 years of age. The estimated SPAP was around 60–70 mmHg measured by TTE. Sildenafil was administered, and his clinical condition improved. During the initial 3 years of follow-up, the estimated SPAP was around 40–60 mmHg, and the levels of BNP were within the normal range. However, clinical deterioration was noted beginning in the 4th year of follow-up. The 6MWT was around 250–300 m. In addition, the levels of BNP elevated gradually. As a result, bosentan was prescribed. Nevertheless, the SPAP still increased gradually to 90–110 mmHg, and the 6MWT declined to 200–250 m. The TAPSE was above 20 mm. The levels of BNP in the most recent years were usually above 300 ng/L. The last RHC during the 9th year of follow-up disclosed high PVR (9.1 WU) and PVRI (17.9 WU.m^2^). A pulmonary scan revealed that there were V/Q match defects in the left lower lobe as well as heterogeneously reduced perfusion in the right upper lobe and right middle lobe. The TTE and chest CT both showed thrombosis on proximal LPA despite treatment with warfarin. The patient did not have an episode of ischemic stroke. Hyperuricemia was found in the same year of follow-up, and allopurinol was administered accordingly. This patient began to take amiodarone in the 11th year of follow-up due to atrial fibrillation. His current NYHA functional status is class III. SpO2 is around 90–95%. He has been followed for 13 years and is currently under treatment with both sildenafil and bosentan.

## 4. Discussion

ES is the most severe phenotype of PAH with multisystem involvement. It is categorized as group 1 pulmonary hypertension according to the 6th World Symposium on Pulmonary Hypertension in 2018 [10,11]. The prevalence of ES in PAH-CHD patients was reported to be between 30 and 40 % [12,13]. The long-term prognosis of patients with ES remains unsatisfactory even under the treatment of advanced therapies. The 10-year survival rate was only 57% [14]. The leading cause of death was heart failure, which accounted for 34.3%, followed by infection and sudden cardiac death [15]. In our study, five out of six patients died during follow-up. Among them, three (60%) died of irreversible right heart failure, and one each died of brain abscess and sudden cardiac death. Although previous studies demonstrated that 5-year survival rates were around 74–83% [14,16,17], no patient mortality was seen within 5 years of follow-up in our studies. The life span of the mortality cases was 53.3 ± 8.2 years, similar to the previous reports [15].

Cyanosis is the major clinical presentation of ES related to hypoxemia, which is attributed to secondary erythrocytosis and polycythemia [18]. It has been shown that erythrocytosis can lead to blood hyperviscosity, which may increase the risk of thrombosis. In addition, patients with ES are predisposed to have coagulation abnormalities which can cause hemoptysis and thrombosis in pulmonary arteries [1]. A number of studies reported that 21–71% of ES patients had intrapulmonary thrombosis [19,20,21]. In our study, four ES patients (80%) had intrapulmonary thrombosis despite warfarin treatment. Taken these results together, the possible etiologies of PA thrombosis in patients with ES included local vascular injury related to high PA pressure, hypercoagulation secondary to cyanosis-related factor deficiencies or platelet dysfunction, red blood cell aggregation caused by sluggish flow in PA, or other embolic sources not related to the formation of thrombus in situ [22]. Some studies revealed that there were no definite benefits from routine anticoagulants on the survival rate for ES patients [14,23]. On the other hand, published guidelines suggest that anticoagulants could be used for atrial arrhythmia or PA thrombus in ES patients [24]. In our practice, we started warfarin according to the guidelines and titrated the dosage to the suggested international normalized ratio of around 1.5 due to a higher risk of intracranial hemorrhage in Asians [25,26]. Another issue found in our study was cerebrovascular infarction associated with paradoxical embolism, especially atrial arrhythmia [27]. Two of our patients had been attacked by an ischemic stroke.

In addition to the aforementioned findings, hyperuricemia has been shown in ES patients with hypoxemia [28]. Up to 50% of patients with ES had hyperuricemia, and the incidence of gouty arthritis was 20% [29]. A retrospective study demonstrated that elevated serum uric acid correlated to the hemodynamic severity of ES, and it was independently associated with long-term mortality [30]. During follow-up, we regularly checked the serum uric acid and administered medication for hyperuricemia in a timely fashion. As a result, all our patients had no gouty arthritis clinically.

Previous studies suggested that progressive RV failure was closely related to mortality in ES patients, and it was associated with RV remodeling and metabolic alterations that could augment glucose uptake [31]. Other studies also disclosed that PAP was correlated significantly with the RV standardized uptake value (SUV) of glucose and the RV/LV ratio in PAH [32,33]. The ratio of RV/LV SUV was shown to be less than 0.5 in healthy people, whereas it could rise to above 1 in patients with PAH [34]. Consistent with these data, our limited patients also demonstrated increased 2-deoxy-2-(^18^F)fluoro-D-glucose (FDG) SUV of RV (Figure 4F), and the ratios of RV/LV were all above 1 (Table 1). Although the clinical outcomes improved after advanced therapies, long-term mortality and morbidity remain high in patients with ES. Due to systemic involvement, multidisciplinary care is warranted. Based on our experience, we regularly followed up with these patients every 1 to 2 months and applied management according to the guidelines of the European Society of Cardiology and American Heart Association [1,35,36]. Similar to some reports [1], we found that the baseline characteristics, including advanced age, SpO2 and WHO functional class, 6 min walk distance (6MWT), brain natriuretic peptide level and systolic pulmonary arterial pressure, could be independent predictors of mortality and closely related to clinical deterioration. In addition, our health care team includes cardiologists, nephrologists, hematologists, endocrinologists, dietitians, and psychiatrists. During the long-term therapeutic strategy, sildenafil was given as first-line treatment, except for case 4 [37,38]; later, the add-on was treated based on individualized need. Although genetic association with PAH has been proposed [5,39], none of our patients received genetic tests. Despite a small patient number, our 5-year mortality rate was zero. However, two of our patients (40%) died during the 6th to 10th year of follow-up. Therefore, the cooperation of multidisciplinary team and other PAH-specific therapies are warranted. The manifestation of metabolism changes disclosed by FDG PET may be a promising target for drug development, especially on pressure-overloaded right ventricles, in the future.

## 5. Conclusions

ES is a severe multisystem disease that requires multidisciplinary care and regular follow-up. Although the mortality rate improved under advanced therapies, the observed clinical complications affect patients’ quality of life and remain challenging for the care team. The collaborative work from our multidisciplinary team, as well as our regular follow-up studies, improved the 5-year survival of our patients. However, further research on other promising therapeutic targets is warranted for the management of ES. Specifically, the characteristics and treatments of ES from other adult groups of PAH, such as idiopathic PAH and PAH associated with connective tissue disease, need to be further explored.

## Figures and Tables

**Figure 1 children-09-01217-f001:**
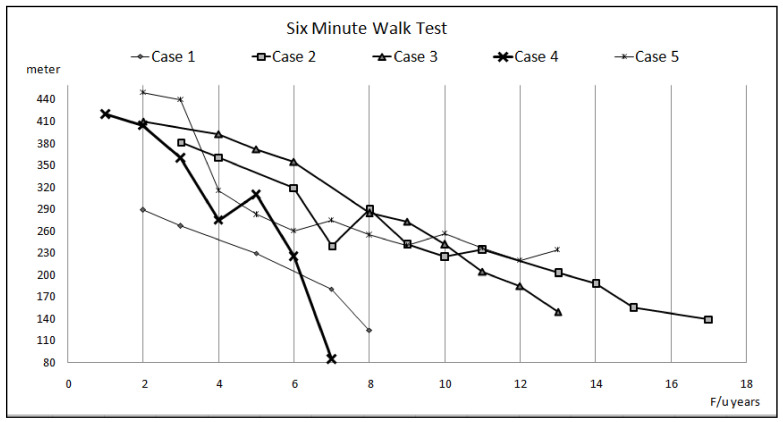
The distance of 6-min walk test in five cases with Eisenmenger syndrome during follow-up.

**Figure 2 children-09-01217-f002:**
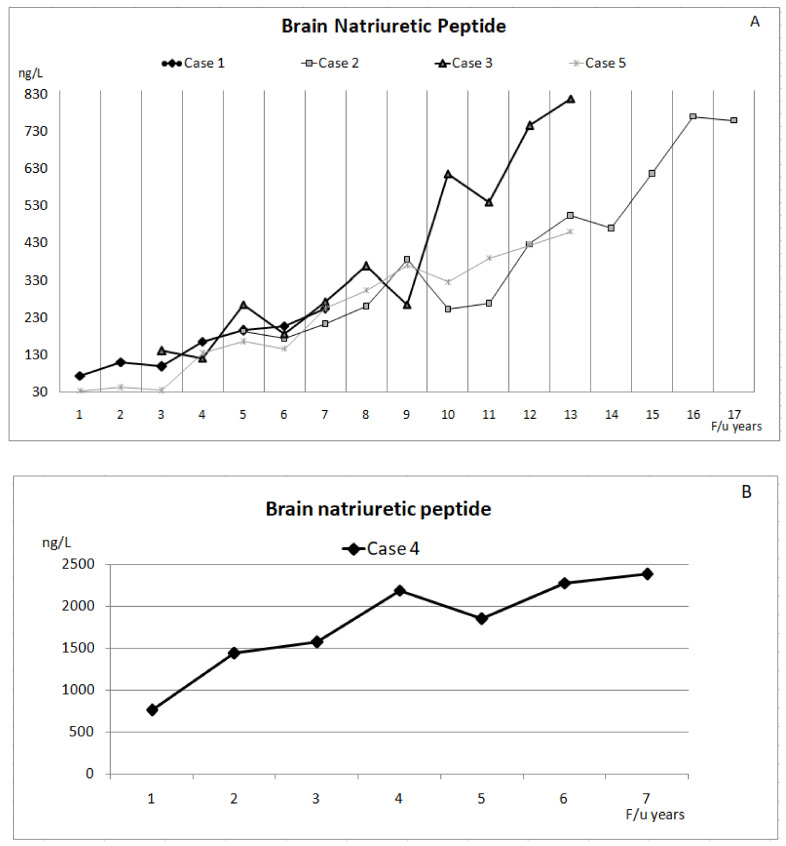
The levels of serum brain natriuretic peptide in case 1, case 2, case 5 (**A**), and case 4 (**B**) with Eisenmenger syndrome during follow-up.

**Figure 3 children-09-01217-f003:**
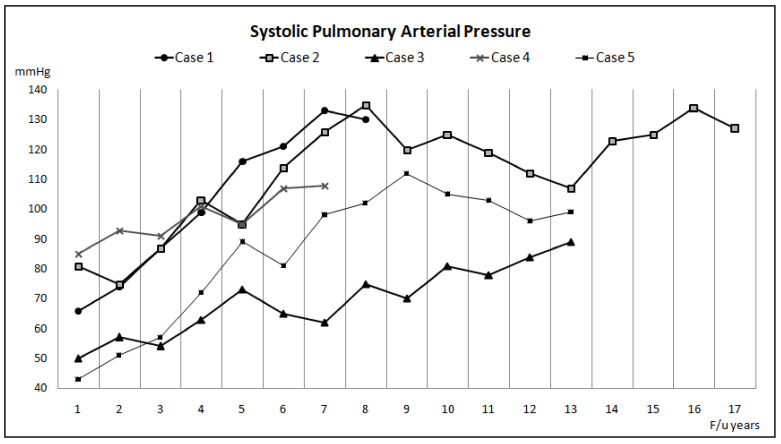
The systemic pulmonary arterial pressure estimated by transthoracic echocardiography in five cases with Eisenmenger syndrome during follow-up.

**Figure 4 children-09-01217-f004:**
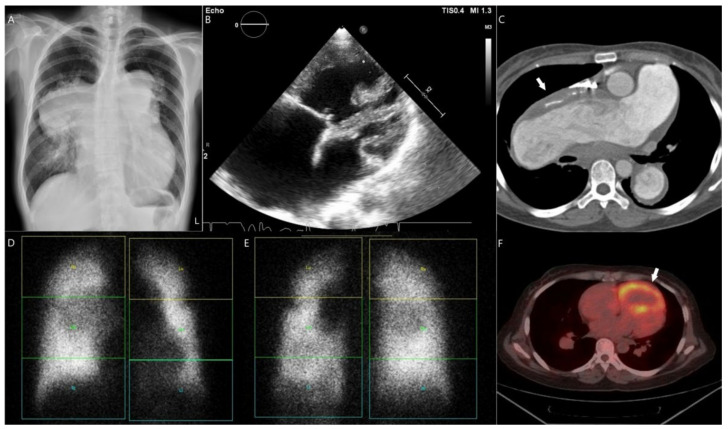
Image findings in a 54-year-old female with Eisenmenger syndrome associated with secundum-type atrial septal defect (Case 2). (**A**) Chest X-ray demonstrated cardiomegaly with a cardiac thoracic ratio of 61% and engorged bilateral pulmonary trunks. (**B**) Four-chamber view of transthoracic echocardiography revealed dilated right atrium and right ventricle. (**C**) Transverse section of chest computed tomography (CT) disclosed thrombus (arrow) on proximal right pulmonary artery. Pulmonary perfusion scan demonstrated defects noted over left lower lobe in anterior view (**D**) and right middle lobe in posterior view (**E**). (**F**) Chest computed scan identified a thrombus located in proximal right pulmonary artery (arrows). The transverse view of positron emission tomography with computed tomography showed prominent fluoro-D-glucose uptake in the right ventricle (arrow) compared to the left ventricle.

**Figure 5 children-09-01217-f005:**
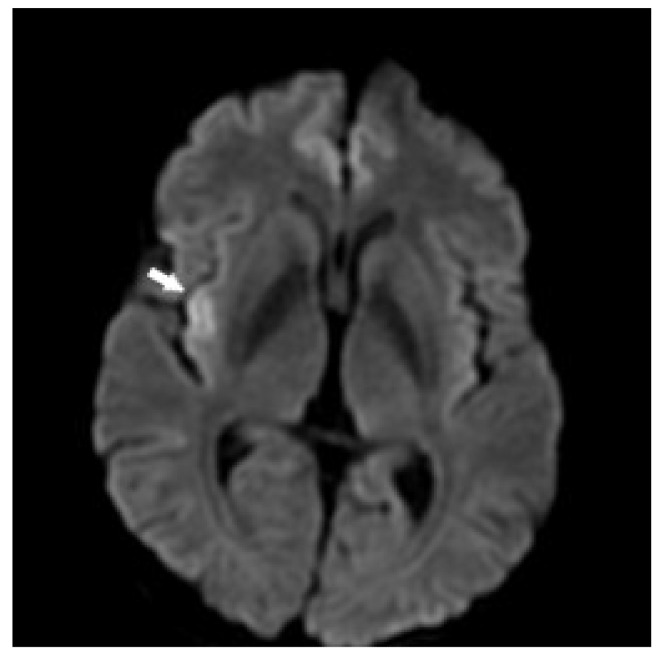
Brain magnetic resonance imaging performed on a 48-year-old female with Eisenmenger syndrome associated with primum-type ASD. The brain magnetic resonance imaging demonstrated high signal intensity at the right insular cortex (arrow) on a transverse view of diffusion-weighted imaging (Case 3).

**Table 1 children-09-01217-t001:** The demographic, clinical, and positron emission tomographic characteristics of patients with Eisenmenger syndrome.

No	Sex	Dx	PET/CT		Tx	Complication				F/u(Yr)
			RV (SUV)	RV/LV		Thrombus In PA	AF	UA	Others	
1	M	VSDII			S, B, I	Bil	-	I	Stroke	8
2	F	ASDII	12.7	1.65	S, B, I	Bil	+	I	-	17
3	F	ASDI	6.2	1.44	S, B, I	RPA	+	N	Stroke	13
4	F	ASDII	5.8	2.15	S, B, I	None	-	I	-	7
5	M	ASDII	12.7	1.31	S, B	LPA	+	I	-	13

Five adults were diagnosed with ES secondary to CHD and were treated and followed in our institution between January 2010 and December 2019. Among them, four were diagnosed and were already subjected to follow-up studies prior to January 2010. Abbreviation: Dx, diagnosis; VSDII, perimembranous-type ventricular septal defect; ASDII, secundum-type atrial septal defect; ASDI, primum-type atrial septal defect; PET/CT, positron emission tomography/computed scan; RV, right ventricle; SUV, standardized uptake value; LV, left ventricle; TX, treatment; S, sildenafil; B, bosentan; I, inhaled Iloprost; PA, pulmonary artery; Bil, bilateral; RPA, right pulmonary artery; LPA, left pulmonary artery; AF, atrial fibrillation; UA, serum level of uric acid; I, increased; N, normal; F/u, follow-up.

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
