# Peer review of "Long-Term Study on Therapeutic Strategy for Treatment of Eisenmenger Syndrome Patients: A Case Series Study"

_children, 2022, doi:10.3390/children9081217_

Round 1

Reviewer 1 Report

In the manuscript entitled “Long-term study on therapeutic strategy for treatment of Eisenmenger Syndrome patients – Case series study” the authors present six cases of patient with Eisenmenger Syndrome including clinical evaluation, therapy, and outcomes. 

My primary concern is regarding patient selection. Specifically, two patients (Case 4 and 5) are described as having repaired congenital heart disease and another patient (Case 6) has Fontan physiology with protein losing enteropathy.  Patients with repaired congenital heart disease associated PAH are separate entity from Eisenmenger Syndrome. It is possible that the two patients with repaired ASD had Eisenmenger Syndrome at diagnosis, but then it is unclear why they underwent surgical repair. Similarly the patient with Fontan failure, protein losing enteropathy, and pulmonary vascular disease is not typically classified as Eisenmenger Syndrome and the authors should consider excluding this patient from the report.

Minor comments.

Did any patients have genetic testing? In particular ASD has been associated with SOX-17 mutation.

Can the authors comment on bosentan was used first line in Case 4? Is there a standard practice for initial and add on therapy for ES in their institution?

Line 88: ATs should be defined.

Line 138: …despite of being treated with warfarin. Edit to remove “of”.

Line 142: Despite of treatment with… Edit to remove “of”.

Table 1 is cutoff and I can not evaluate this table. Please edit for formatting. 

The authors could consider creating a figure showing change in parameters over time for each patient (pulmonary artery pressure, PVR, BNP, six minute walk distance). This would add significantly to the manuscript. 

Author Response

Dear Reviewer 1:

Firstly, we deeply appreciate the help and valuable comments. The replies will be as follows

Major Comment:

My primary concern is regarding patient selection. Specifically, two patients (Case 4 and 5) are described as having repaired congenital heart disease and another patient (Case 6) has Fontan physiology with protein losing enteropathy.  Patients with repaired congenital heart disease associated PAH are separate entity from Eisenmenger Syndrome. It is possible that the two patients with repaired ASD had Eisenmenger Syndrome at diagnosis, but then it is unclear why they underwent surgical repair. Similarly the patient with Fontan failure, protein losing enteropathy, and pulmonary vascular disease is not typically classified as Eisenmenger Syndrome and the authors should consider excluding this patient from the report.

Reply:

Indeed, the diameters of atrial septal defect in Case 4 and 5 were more than 2 centimeters with advanced pulmonary hypertension before operation. In fact, the repair of cardiac defects didn’t improve their situation and the condition of these two patients deteriorated after surgery. Then, both of them were referred to our tertiary hospital by cardiac surgeons.

We excluded the Case 6 with Fontan failure from our study. Thank you for the suggestion.

Minor comment:

#1. Did any patients have genetic testing? In particular ASD has been associated with SOX-17 mutation.

Reply:  All of our patients didn’t receive the genetic testing. We add some discussion with genetic test in the last paragraph of Discussion (Page 9, line 308-309).

#2. Can the authors comment on bosentan was used first line in Case 4? Is there a standard practice for initial and add on therapy for ES in their institution?

Reply:  In our institute, we used the sildenafil as the first line agent recently. In fact, sildenafil and bosentan were not covered by the national health insurance in the beginning. In Case 4, she fortunately had a chance to receive compassionate use with bosentan at initial treatment. As a result, she received bosentan initially instead of sildenafil. Accordingly, we treated the ES patients based on the guidelines of European Society of Cardiology and American Heart Association from. Therefore, in 9 page, from 303 to 305 line ,it was written as " and applied management according to the guidelines of European Society of Cardiology and American Heart Association[35-37]."

#3.Line 88: ATs should be defined.

Reply:  We made the complementary definition of ATs in the second paragraph of Introduction (Page 2, line 60-61).

#4. Line 138: …despite of being treated with warfarin. Edit to remove “of”. Line 142: Despite of treatment with… Edit to remove “of”.

Reply: We removed ‘’of’’ in page 6 , line 167 and line 172.

#5. Table 1 is cutoff and I can not evaluate this table. Please edit for formatting. 

Reply: We reformatted table 1 clearly, instead of cutoff in page 3 line 93-95. Thank you

#6. The authors could consider creating a figure showing change in parameters over time for each patient (pulmonary artery pressure, PVR, BNP, six minute walk distance).

Reply:  We had created a new figure about changes in parameters including pulmonary arterial pressure, BNP, and six minute walk distance, respectively.

Sincerely

Zen-Kong Dai,  MD, PhD,
Professor, Pediatric Cardiology and Pulmonology
 Kaohsiung Medical University.

Kaohsiung, Taiwan

[email protected]

886-7-3121101

Reviewer 2 Report

retrospective analysis with small volume of patients

Although there is no risk stratification model  for eisenmenger, i would prefer oxygen saturation, RA area, pericardial effusion, TAPSE, CI, RAP instead of SPAP.

I believe there are no evidence that INR of 1.5 is therapeutic

It is a very good question (i have no asnwer) if we need to treat Eisenmenger with anticoagulation or not

Author Response

Dear Reviewer 2:

Firstly, we deeply appreciate the help and valuable comments. The replies will be as follows

Comments:

#1. Retrospective analysis with small volume of patients.

Reply: We do understand the weakness, and it is therefore written in page 9, line 309 as " Despite a small patient number,"

#2. Although there is no risk stratification model for eisenmenger, i would prefer oxygen saturation, RA area, pericardial effusion, TAPSE, CI, RAP instead of SPAP.

Reply: By sequential analysis, we added the parameters such as oxygen saturation, TAPSE and systolic pulmonary arterial pressure, instead of risk stratification. Only case 4 patient, ASDII, presented with pericardial effusion, with shortest follow-up year in our study. We would like to share our experience for early evaluating the development of right heart failure in these patients.

#3. I believe there are no evidence that INR of 1.5 is therapeutic, It is a very good question (i have no answer) if we need to treat Eisenmenger with anticoagulation or not.

Reply: In the second paragraph of discussion in page 8, line 277-282, we explained that routine administration of anticoagulant in ES is controversial. However, anticoagulant usage is reasonable in patients with concomitant atrial fibrillation or pulmonary thrombosis. The reason of INR around 1.5 is due to higher risk of intracranial hemorrhage is Asian people.

Sincerely

Zen-Kong Dai,  MD, PhD,
Professor, Pediatric Cardiology and Pulmonology
 Kaohsiung Medical University.

Kaohsiung, Taiwan

[email protected]

886-7-3121101

Round 2

Reviewer 1 Report

I would like to thank the authors for their substantive revisions and addressing the majority of my concerns. I have include two additional comments for consideration.

Major Comment:

I continue to be concerned about the inclusion of two patients with repaired congenital heart disease associated PAH, which are a separate entity from Eisenmenger Syndrome. While the authors note the patients did not clinically improve after surgical repair, it is difficult to know if they would have done better without surgical repair. While my opinion is these patients have different physiology then the other Eisenmenger Syndrome patients presented and confound the results, the authors are entitled to a different opinion. 

Minor Comment: 

1. The authors could consider additional reference of Figures 1-3 and condensing the description of change in individual values.

Author Response

We deeply appreciate the kind and valuable comments. The replies will be as follows

Major Comment:

I continue to be concerned about the inclusion of two patients with repaired congenital heart disease associated PAH, which are a separate entity from Eisenmenger Syndrome. While the authors note the patients did not clinically improve after surgical repair, it is difficult to know if they would have done better without surgical repair. While my opinion is these patients have different physiology then the other Eisenmenger Syndrome patients presented and confound the results, the authors are entitled to a different opinion..

Reply:

We completely agree with you that case 4 and case 5 could be a separate

entity, or different phenotype from the Eisenmenger Syndrome patients without

surgical repair. Some authors reported that the Eisenmenger physiology is part

of the spectrum of unoperated congenital heart disease, but may also develop

despite surgical repair of the underlying cardiac anomaly [Kaemmerer H: Curr

cardiol Rev. 2010,;(4):343-55.]

However, we did not make sure that they could be better without surgical repair, because that there was not well sequential and timing study of pulmonary vascular remodeling and altered expressions of vasomediators such as endothelin-1, endothelial nitric oxide synthase and prostaglandin et al. during the progression of ES. In addition, we agree with you that the pathophysiology of both cases is different from the other ES patient, in which changed hemodynamic status noted from high left to right flow to reversal flow.

Minor comment:

The authors could consider additional reference of Figures 1-3 and condensing the description of change in individual values.

Reply

On page 9, line 305-308, it was rewritten asSimilar to some reports [38], we found that the baseline characterstics including advanced age, SpO2 and WHO functional class, 6 min walk distance (6MWT), brain natriuretic peptide level and systolic pulmonary arterial pressure could be independent predictors of mortality, and closely related to clinical deterioration.”

Sincerely

Zen-Kong Dai, MD, PhD,
Professor, Pediatric Cardiology and Pulmonology
Kaohsiung Medical University.

Kaohsiung, Taiwan

[email protected]

886-7-3121101
